**Subject Category:**
Biology (whole organism)

behaviour/cognition/computational biology

animal song, gibbon, language, Bayesian analysis, context-free grammar

**Authors for correspondence:**
T. Morita
e-mail: tmorita@alum.mit.edu
H. Koda
e-mail: koda.hiroki.7a@kyoto-u.ac.jp

# Superregular grammars do not provide additional explanatory power but allow for a compact analysis of animal song

## T. Morita and H. Koda

Primate Research Institute, Kyoto University, 41-2 Kanrin, Inuyama, Aichi 484-8506, Japan

TM, 0000-0002-3900-5410; HK, 0000-0002-0927-3473

A pervasive belief with regard to the differences between human language and animal vocal sequences (*song*) is that they belong to different classes of computational complexity, with animal song belonging to regular languages, whereas human language is superregular. This argument, however, lacks empirical evidence since superregular analyses of animal song are understudied. The goal of this paper is to perform a superregular analysis of animal song, using data from gibbons as a case study, and demonstrate that a superregular analysis can be effectively used with non-human data. A key finding is that a superregular analysis does not increase explanatory power but rather provides for compact analysis: fewer grammatical rules are necessary once superregularity is allowed. This pattern is analogous to a previous computational analysis of human language, and accordingly, the null hypothesis, that human language and animal song are governed by the same type of grammatical systems, cannot be rejected.

## 1. Introduction

Language is often considered a property unique to humans, in contrast with other animal behaviour such as birdsong [1–3]. It is a commonly held belief that human language and animal vocal sequences (which we term *song* herein) belong to different classes of computational complexity. Animal song belongs to the class of regular languages and is modelled by regular expressions, finite-state automata and left/right-branching grammars. Human language, on the other hand, requires superregular analyses. This argument, however, is not supported by empirical evidence, since superregular analyses of animal song are significantly understudied. In particular, only

**Table 1.** Log prior, likelihood and posterior (minus the normalizing constant) probabilities of PCFG and the regular grammar reported in [16]. Scores of the best grammars ('CFG-L' for PCFG and 'REG-B' for regular) are cited.

| probability type | PCFG | regular |
|---|---|---|
| log prior | **−1111** | −1943 |
| log likelihood | −25 889 | **−25 368** |
| total = log posterior (−normalizer) | **−27 000** | −27 311 |

regular analyses have been performed in the original studies on animal song cited by proponents of the argument [1–3]: e.g. song of Bengalese finch—the most popular example discussed in the literature—has been analysed with $n$-gram models [4], $k$-reversible finite-state automata [5] and hidden Markov models [6], which are all regular and no study has yet assessed superregular models. The goal of this paper is to perform a superregular analysis of animal song and demonstrate that it can be applied to realistic non-human data. Song data from gibbons were used, since the gibbon is an ape known for its long vocal sequences [7–9]. This superregular analysis does not increase explanatory power but, instead, provides a compact analysis, since fewer grammatical rules are necessary once superregularity is included. This is in accordance with the previous computational analysis of human language that is reviewed in the next section. Given this, the null hypothesis that human language and animal song are governed by the same kind of grammatical systems cannot be rejected.

## 2. Preliminaries

A classic argument for the superregularity of human language is based on a mathematical result from formal language theory. This argument assumes that human language allows an unbounded depth of the centre-embedding schematized as $N^m V^m$, where N and V stand for noun and verb (phrases), respectively, and $m \in \mathbb{N}$ represents the number of repeats. For example, *The mouse died.* ($m = 1$), *The mouse [the cat chased] died.* ($m = 2$), *The mouse [the cat [the dog bit] chased] died.* ($m = 3$), …. Under the assumption of unbounded centre-embedding, human language is non-regular [10,11]. However, this is empirically unsupported since the depth of centre-embedding observed in corpora is $m \leq 3$ [12] and sentences with multiple centre-embedded clauses are extremely hard for people to process or accept [13–15]. These empirical results imply that we would have little chance of observing centre-embedding in animal data even if the system behind the data has a superregular architecture.

A better empirical argument for the superregularity of human language—in the sense that the approach is more easily applicable to animal studies—comes from probabilistic analysis. Perfors *et al.* [16] showed that probabilistic context-free grammar (PCFG), which is a superregular language model, is more probable for the analysis of child-directed speech than regular grammars in the posterior. This finding is significant as it provides an important advantage of PCFG analysis, which becomes visible when we decompose the posterior probability. Using Bayes' Theorem, it is found that the posterior probability $p(g \mid d)$ of a grammar $g$ given data $d$ is proportional to the product of the prior probability $p(g)$ of $g$ and the likelihood $p(d \mid g)$ of $d$ given $g$. Thus, taking the log of the probabilities:

$$\log p(g \mid d) = \log p(g) + \log p(d \mid g) - \log p(d) \quad (-\log p(d) \text{ is the normalizing constant.})$$

The likelihood encodes the grammars' explanatory power of the data. The prior, on the other hand, evaluates the compactness of the grammars, with smaller grammars being more probable. Table 1 shows the log prior, likelihood and posterior (minus the constant normalizer term; i.e. the sum of the log prior and likelihood) of the best PCFG and regular grammar found by Perfors *et al.* The advantage of the PCFG over the regular grammar in the posterior comes from the prior, not the likelihood. In other words, the empirical advantage of a PCFG analysis of human language is not that it can explain more data than regular grammars, even though hypothetical data exist that can be explained only by superregular grammars (unbounded centre-embeddings). Instead, PCFG is a more compact analysis of human language than regular grammars.

## 3. Overview of the material and methods

In this section, we outline our material and methods. Technical details are available later in the Material and methods section.

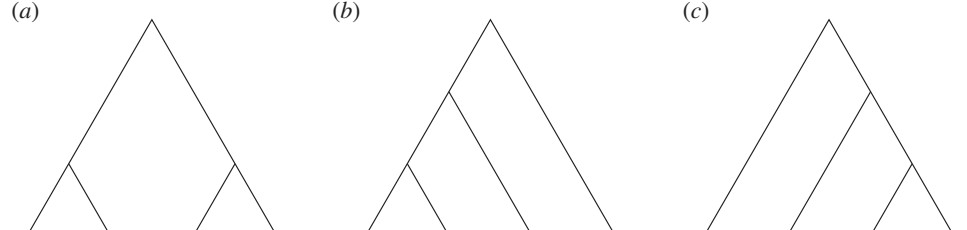

**Figure 1.** Possible parses of a string of length 4. The terminal vertices at the bottom of the tree graphs correspond to the individual data points of the string. (*a*) Non-regular, (*b*) left-branching (regular) and (*c*) right-branching (regular).

The data subject to the grammatical analysis are sequences of acoustic feature vectors (mel-frequency cepstrum coefficients), measured at representative data points of gibbon song recordings. This data format is different from outputs of the standard PCFG: a PCFG outputs string of discrete symbols (called *terminals*), such as words in human language syntax. Accordingly, we extend the standard PCFG with multivariate Gaussian emission from terminals (cf. [17]). To generate a data sequence, our model first generates a string of terminals using PCFG. Then, a Euclidean vector is generated according to a multivariate Gaussian conditioned on each of the terminals. Our analysis can be viewed as a joint inference of (i) the discrete categories—corresponding to the terminals—of the acoustic feature tokens and (ii) the grammar behind the sequential patterns of the categories. While these two components have been analysed in separate steps in the previous studies on animal song [4–6,18,19], simulation studies of human language learning have shown that a joint inference of multiple aspects of the target language—capturing their correlations—is more successful than separate learning of the components [20–23]. Hence, we consider that the joint analysis is more reliable than the previous two-step analysis.

A PCFG consists of grammatical production rules associated with probability of their use. A challenge in a PCFG analysis of animal data is that we do not know a particular set of grammatical rules appropriate for the data as well as probability of the rules. In the current study, we take a Bayesian inference approach to this problem, estimating the posterior probability distribution over PCFGs (both rules and their probability) conditioned on the gibbon data. Importantly, regular grammars are a special case of PCFG that only generate either the left-branching (figure 1*b*) or right-branching (figure 1*c*) structures. Hence, the posterior tells us how probable regular grammars are among PCFGs given the gibbon data.

There are infinitely many sets of context-free grammatical rules, and thus it is not appropriate to assume the uniform prior over all the possible PCFGs (improperness). This study adopts the hierarchical Dirichlet process (HDP) for the prior over PCFGs [24,25]. The HDP prior introduces a bias for *compact* PCFGs: it assigns exponentially smaller prior probability to PCFGs whose production probability mass are spread over a greater number of grammatical rules, favouring those with a smaller number of reusable rules. The Bayesian inference balances the PCFG likelihood (fit to the data) and the HDP prior (compactness), and the posterior probable PCFGs are those that can generate the data with high probability while reusing a limited number of production rules. Similar balancing between the explanatory power (likelihood) and compactness (prior) is widely adopted in scientific evaluation of models [26–28] as well as modern theories of learning [16,29–31].

In practice, it is difficult to directly measure the posterior probability of the regular and non-regular grammars (even with the help of approximation of the posterior). Accordingly, we will report the expected counts of regular and non-regular parses of the training data and held-out test data based on the posterior. The logic is as follows: if the regular grammars are the only probable accounts of the gibbon data in the posterior, then only the regular parses of the data would have large expected counts. We will show that the expected counts of non-regular parses are not small at all in reality (in comparison with the counts of regular parses as well as a random baseline), and thus non-regular grammars are unignorably probable analyses of the gibbon data, countering the previous argument for their regularity [1–3].

The analysis outlined above will show that non-regular grammars are probable analyses of the gibbon data. However, it would not tell us what made the non-regular grammars probable: (i) Did the non-regular grammars enable better fit to the data (explanatory power)?; (ii) Did the non-regular grammars allow for a more compact explanation of the data, using a more limited variety of production rules (compactness)? To address these questions, we will further diagnose the induced

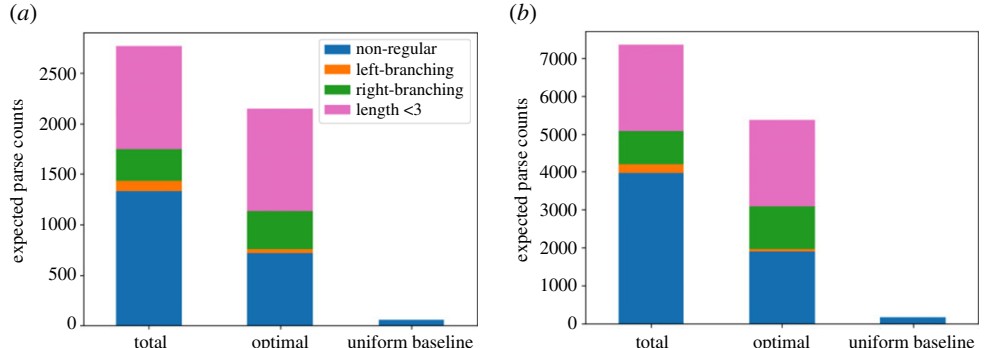

**Figure 2.** Expected counts of parses of the training (*a*) and test (*b*) data. The 'total' bar shows the total expected counts of each parse type. The 'optimal' bar shows the expected counts of the optimal parses. The 'uniform baseline' bar shows the expected counts of the non-regular parses under the assumption of the uniform distribution over the logically possible parses of each data string.

posterior inference of PCFGs by comparing it with that of hidden Markov models (HMMs) induced from the same data and the HDP prior [24,32]. HMMs restrict possible grammars to the regular ones, and the comparison helps us understand what kinds of contribution the non-regular grammars—only possible when the hypothesis space is the PCFGs—make to the posterior inference. The question about explanatory power of the non-regular grammars (i) is addressed by comparing the PCFG- and HMM-based posterior predictive probability of the held-out test data. The compactness of grammars (ii) is evaluated by the expected type counts of production rules used for a PCFG/HMM to generate the training data.

# 4. Results

## 4.1. Posterior inference of PCFG

The 'total' bar of figure 2*a* shows the expected counts of left-branching, right-branching and non-regular parses of the training data. The parse of the data strings shorter than 3 is unique and regular (both left- and right-branching), and their counts are reported separately (length < 3). A total of 48.00% of the training data (1328.08 strings of the total $N = 2767$) are expected to have non-regular parses. The ratio increases to 86.97% when we focus on the strings of a length of at least $4(N = 1527)$, for which non-regular parses are logically possible. Note that these large proportions are not due to the broader variations of non-regular parses than regular parses: in other words, it is not the case that all the possible parses are almost uniformly probable (i.e. induction failure) and the $\frac{1}{l}\binom{2(l-1)}{(l-1)} - 2$ non-regular parses of each string (whose length is represented by $l$) ganged up against the only two regular parses. We can see this from the 'optimal' bar of figure 2*a*, which shows the expected counts of the Bayes-optimal parses (with the greatest posterior probability) [25]. The expected counts of the optimal parses tell us how confidently the data were parsed: e.g. if a data string has a unique probable parse (=optimal parse), the expected count of that parse is almost 1. If all the possible parses of the data string are equally probable, on the other hand, the expected count of the optimal parse is only $l\binom{2(l-1)}{(l-1)}^{-1}$. Figure 2*a* shows that the expected counts of the non-regular optimal parses of the training data are 717.35 and smaller than the total expected counts of the non-regular parses—meaning that there was some uncertainty in the parsing. However, the expected counts of the non-regular optimal parses are still an order of magnitude greater than the uniform baseline (56.11, represented by the 'uniform baseline' bar of figure 2*a*). The expected counts of the non-regular optimal parses are also greater than those of the left-branching parses (105.92 in total, 37.80 for the optimal) and the right-branching parses (315.00 in total, 378.33 for the optimal).

A similar pattern was observed with the held test data (figure 2*b*). A total of 54.13% of the entire data (3974.86 strings of the total $N = 7343$) and 88.23% of the strings of a length of at least 4 ($N = 4505$) are expected to have non-regular parses. The expected counts of the non-regular optimal parses are 1895.36, an order of magnitude greater than the uniform baseline (166.23). The expected counts of the non-regular optimal parses are also greater than those of the left-branching parses (217.72 in total, 61.51 for the optimal) and the right-branching parses (886.42 in total, 1144.21 for the optimal).

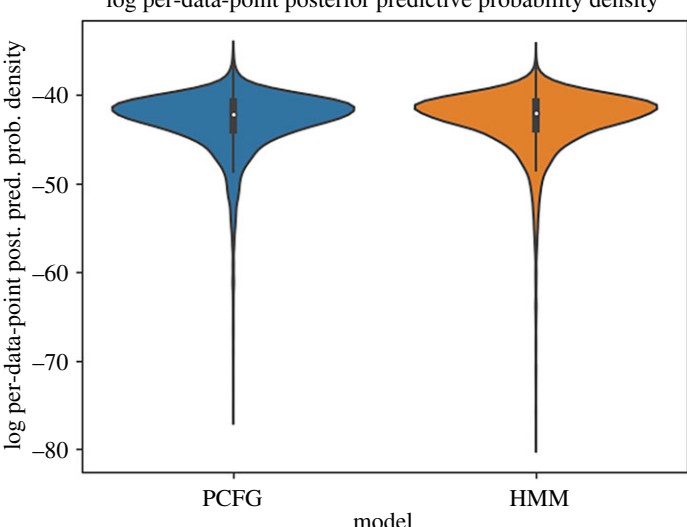

**Figure 3.** Log per-data-point posterior predictive probability density of test data ($N = 7343$) with PCFG and HMM.

**Table 2.** Expected type counts of production rules used for a PCFG/HMM to generate training data. The type counts were calculated in two ways by applying the step function $\mathbb{1}\,[\,\cdot\geq 1\,]$ and $\tanh(\cdot)$ to the expected token frequency of rules.

| activation function | HMM | PCFG |
|---|---|---|
| $\geq 1$ | 274 | **222** |
| tanh | 307.523023 | **230.429057** |

In summary, the results suggest that a large portion of the gibbon data was parsed in non-regular ways, and thus non-regular grammars are unignorably probable analyses of the data in the posterior.

## 4.2. Predictive power

Figure 3 shows the distribution of the log posterior predictive probability density of the test data divided by the string lengths (and thus 'per data point') with the induced PCFG (mean: $-42.900044$, std: $3.421662$) and HMM (mean: $-42.797174$, std: $3.535390$). The mean density of the PCFG is slightly smaller than that of the HMM. This generalization holds even when we focus on the long-string portion of the data, whose length is 4 or greater and non-regular parses are possible ($N = 4505$; mean: $-41.954381$ and std: $1.886496$ with PCFG; mean: $-41.794641$ and std: $1.964568$ with HMM). This indicates that the non-regular parses allowed by the PCFG did not provide extra predictive power beyond the regular parses produced by the HMM.

## 4.3. Compactness

Table 2 reports the expected type counts of production rules used for a PCFG/HMM to generate the training data: PCFG is expected to generate training data with 52 (counted with $\mathbb{1}\,[\,\cdot\geq 1\,]$) or 77 (with tanh) fewer rules and thus allows for a more compact analysis than HMM.

## 5. Discussion and conclusion

The parse induction result shows that non-regular parses were more probable latent structures of gibbon song than the regular parses. This implies a superregular strong generative capacity of the gibbon's grammar, which contradicts the previous opinion that regular grammars are sufficient to analyse animal song [1–3]. We neither claim that the gibbon is an exceptional species that exhibits a superregular system (due to, for example, its phylogenetic closeness to humans compared to songbirds) nor that the superregular versus regular boundary is contiguous between primates and

non-primates. Superregular analyses of animal song are currently understudied, and thus, it is an open question whether such analyses are appropriate for other specific species.

Given that animal song and human language both deserve a superregular analysis, their comparative studies could be more meaningful for our understanding of human language evolution than previously claimed by theoretical linguists [3]. The skepticism on the effectiveness of such comparative studies was based on the assumption that animal song was regular and thus there was no hope to find grammatical similarities between human language and animal song. By analysing human language and animal data with the same class of grammars, we would obtain a deeper understanding of their similarities and differences, which could in turn help us build a more sophisticated theory of human language evolution.

Another key finding in the present study is that PCFG does not improve fit to the gibbon data in comparison with HMM. Instead, the advantage of the superregular analysis of gibbon song is its compactness: fewer rules are necessary to analyse the data if non-regular parses are allowed. Note that this pattern is consistent with previous observations of human syntax: a PCFG is more probable than HMMs/regular grammars given human language sentences, because it reduces the grammar size and thus enables greater prior probability, even though its predictive power (i.e. likelihood) is smaller [16]. The importance of the compactness metric in grammar evaluation has long been emphasized in the field of theoretical linguistics (by the proponents of the regular versus superregular difference between animal song and human language) [33,34]. However, the previous studies of animal song have evaluated models solely by the predictive power of the data [6,19], and the compactness metric has been overlooked. For future studies on animal song syntax, it is important to remember that the advantages of PCFG and other superregular hypotheses may be expressed in the form of compactness/greater prior, rather than improved model explanatory power or likelihood. Bayesian inference and similar approaches (e.g. minimum description length [28,31]) combine the two metrics in a mathematically natural way.

Finally, note that the (posterior distribution of) PCFG induced here is not guaranteed to match the actual system behind gibbon song. The induced PCFG is just a computationally optimal hypothesis for the gibbon data at this point, and its biological validity should be assessed in future experimental studies (e.g. callback experiment). Moreover, our hypothesis space was still limited to the class of PCFGs, whereas the actual gibbon grammar might go beyond the context-free generative capacity. Sticking to the same PCFG hypothesis space in future studies would lead to the same error as the previous assumption of regularity. Importantly, linguists have suggested that (P)CFG is insufficient to fully describe human language [35,36], and that at least mildly context-sensitive computational power is necessary [37]. The recent success of language models based on recurrent neural networks further implies that some aspects of linguistic data are better captured by a more powerful, Turing-complete architecture. PCFG itself has also been improved so that it better captures frequently recurring structural patterns [38]. We—researchers of animal song—should open our eyes to these more recent achievements in computational and theoretical linguistics and adopt the new analytical techniques to make the comparative studies more fruitful.

# 6. Material and methods

## 6.1. Data

We recorded songs of three male gibbons in captivity at the Primate Research Institute, Kyoto University. One was an agile gibbon (H. agilis, age: 44 year or older), and the others were hybrids of H. agilis and H. albibarbis (age: 19 and 18 year). (Note that H. albibarbis was long considered a subspecies of H. agilis [7,39] until a recent DNA study [40], and thus, the two species are phylogenetically similar.) The songs were originally recorded on an eight-channel microphone array (TAMAGO-03, System in Frontier, Inc., with the FPGA customized such that −12 dBSPL of additional gain was applied), and we used the recordings' first channels for the analysis. The training data were recorded between 10 and 19 August 2017, between 18.00 and 9:00. The test data were recorded between 1 and 30 September 2017, during the same morning periods. The sampling rate was 16 000 Hz.

The multi-channel recordings were intended for use in localization of the sound sources (i.e. singer identification) and separation of sounds from different sources, but neither the localization nor separation was successful in our recording environment (we used the HARK programs with the default and customized transfer functions [41]). Nevertheless, previous observations of wild gibbons suggest our recorded songs would have overlapped little among the three singers: it has been

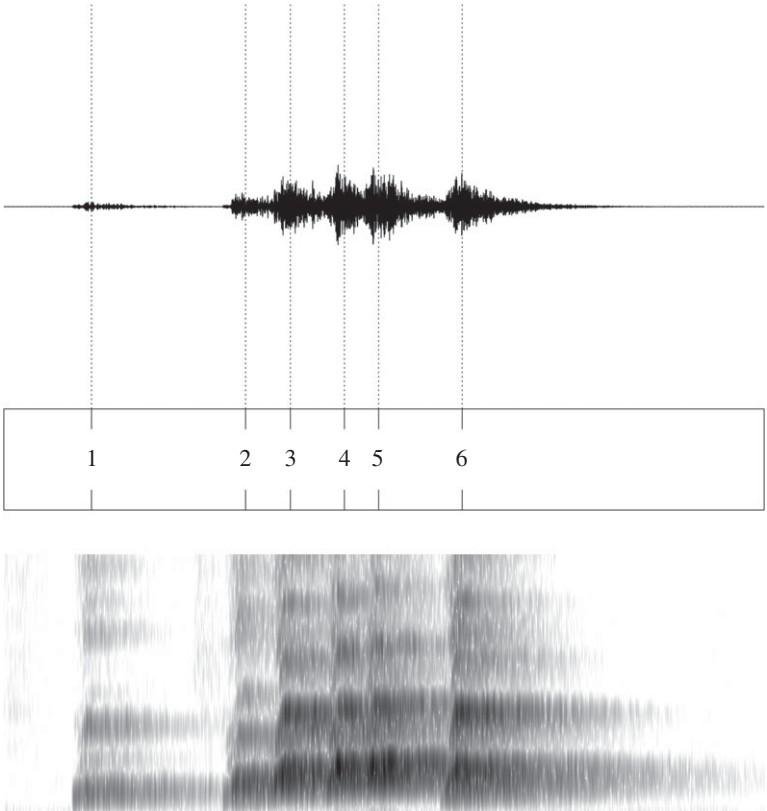

**Figure 4.** Local peaks in a single song region. The top and bottom rows show the raw sound wave and the spectrogram containing the song region, respectively. The vertical dotted lines stretching from the middle row to the sound wave represent the local peaks detected by the algorithm.

reported that males typically vocalized in turn, with those in adjacent territories staying silent during another's singing [42–44]. The rarity of overlaps is also in accordance with our impression of the recordings. In addition, we suspect that the existence of overlaps (if any) would not make significant differences in the results of our grammatical analysis reported here. It is easily proved that switching among multiple finite-state automata can be seen as one large finite-state automaton, and thus, data strings produced by such a process constitute a regular language in terms of formal language theory. Although our approach is probabilistic and thus such a claim from formal language theory does not guarantee the validity of our conclusion, the proof above still suggests that data strings produced by multiple individuals do not immediately lead to superregularity. (Note, however, that female gibbons are known to vocalize simultaneously—a type of song called *great calls*, and communication among multiple individuals is of major interest to researchers of the species [8]. Hence, the sound localization and separation are an important issue that should be addressed in future studies on gibbon song.)

## 6.2. Preprocessing

The inputs to our grammatical analyses were strings of acoustic features at representative points in the recorded sound streams. We first detected regions containing gibbon song in the sound streams by calculating the ratio $r$ of the energy in the song-sound band (500–1500 Hz) to that in the total frequency band (0–8000 Hz). Each frequency band's energy was the sum of the squared amplitudes in the band obtained from the sound spectrum calculated by the discrete Fourier transform (DFT-windows: 20 ms, DFT-overlap: 10 ms). A region of length greater than or equal to 500 ms was judged to contain song if $r > 0.6$ at every point in the region.

We then downsampled the song regions, leaving only representative data points whose acoustic features constituted data strings that would be subject to the grammatical analyses. By the definition of the song regions above, there were no identifiable silent intervals inside the regions that could define smaller units of the song akin to birdsong syllables [18]. On the other hand, the song regions often consisted of multiple components similar to human speech syllables (figure 4), each of which

was considered to correspond to an atomic event of vocal production. Accordingly, acoustic features at local peaks of the wideband envelope were used as the unit components of the string data for the grammatical analysis. The local peaks were systematically detected by the recent method using the Hirbert-transformed signals based on the bandpass-filters designed to model human auditory functions [45]. The acoustic local peaks identified by this method are said to correspond to the articulatory local maxima of mouth opening in the production of human speech syllables. Given the anatomical similarities between the vocal tract of the human and the gibbon [46], we considered that the method was appropriate to locate representative data points of the gibbon song.

The details of the local peak detection algorithm are as follows. We first bandpass-filtered waveforms in the song regions into six frequency bands with cut-off frequencies at 80, 260, 600, 1240, 2420, 4650 and 7999 Hz [45,47]. The resulting bandpassed signals were then Hilbert-transformed, and the absolute values of the six transformed signals (narrowband envelopes) were summed to obtain a wideband envelope [45]. The search for the local peaks of the wideband envelope started with the detection of their candidates. Each local peak candidate was the peak within a 300 ms search window slid 10 ms at a time throughout the recordings. The candidates were adopted as local peaks if the following two conditions were met: (i) the difference between the local peak and the minimum in the 300 ms search window was at least 1.0 (Hirbert unit); and (ii) a local peak was not preceded by another one within 150 ms.

Finally, we calculated the 13 Mel-Frequency Cepstrum Coefficients (MFCCs) of the 100 ms window starting from each of the local peaks. While MFCC was originally designed to capture acoustic features of human speech, it has also been adopted in analyses of vocal activities of primates [48] and other species [49]. The present study also followed this standard approach. Note, however, that neural networks have recently started replacing MFCC in the field of computational linguistics [50], and future studies on gibbon song and other animal vocal activities should also consider such new methods, just as we suggested for grammatical analyses of animal data.

The MFCC vectors calculated at the local peaks were clustered into strings by the song regions containing the peak locations. This yielded 2767 strings of training data and 7343 strings of test data. (The reason behind the smaller amount of training data than that of the test data was that the induction of PCFG was time-consuming and the amount adopted here was almost the maximum possible given our limited computational resources. The size of the test data, on the other hand, did not have such restrictions, so we used the greater amount—recordings over a month.) We fed these strings of MFCC vectors to the grammatical analyses described below.

## 6.3. Generative models

We adopted the basic design of PCFG with an HDP prior (HDP-PCFG) discussed in [24,25]. We modified their HDP-PCFG in three ways. First, we assumed that the roots of PCFG parse trees could be variable, instead of assuming a unique node label like Chomsky normal forms. This modification allowed us to analyse incomplete data strings that appear in naturalistic data. For instance, English corpora can involve subjectless sentences (e.g. *Just came back from the party.*) and strings solely consisting of a noun phrase (e.g. *Great job.*). Variable roots enable a uniform treatment of such incomplete strings and substrings of complete strings. For example, *just came back from the party* could be parsed as a verb phrase whether it stands alone or appears within a complete sentence like *John just came back from the party*. Secondly, our data were MFCC vectors (of $D := 13$ dimensions) rather than discrete symbols, and thus, the terminal symbols were latent variables in our HDP-PCFG. Accordingly, we also imposed an HDP prior on the terminal production rules. Finally, we assumed a Gaussian emission of the MFCC vectors conditioned on terminals of the HDP-PCFG, with Gaussian and Inverse Wishart priors on the mean and covariance matrix parameters respectively [23,51,52]. This generative model is formally described in figure 5a,c.

We also followed [24] and imposed an HDP prior on the HMM (HDP-HMM). The only difference was that MFCC vectors, rather than discrete symbols, were emitted conditioned on the hidden states [52]. This model is summarized in figure 5b,c.

The free concentration parameters of the HDPs and beta distributions, denoted by $\alpha$ with subscripts, were all set to 1. The scale matrix of the Inverse Wishart prior of the Gaussian emission was the identity matrix, and the degree of freedom was $\nu_0 = D - 1 + 0.001 = 12.001$. The mean $\mathbf{m}_0$ of the Gaussian prior was set to the sample mean of all the MFCC vectors, and the scalar of the covariance matrix was $k_0 = \nu_0 = 12.001$ (cf. [23]).

(a)

$$\boldsymbol{\beta} \sim \text{GEM}(\alpha_{\text{NonTerm}}) \qquad \text{draw top-level non-terminal probabilities}$$

$$\boldsymbol{\gamma} \sim \text{GEM}(\alpha_{\text{Term}}) \qquad \text{draw top-level terminal weights}$$

$$\phi^{(R)} \mid \boldsymbol{\beta} \sim \text{DP}(\alpha_{\text{RT}}, \boldsymbol{\beta}) \qquad \text{draw root-labeling weights}$$

For each non-terminal $z$,

$$\phi_z^{(T)} \mid z \sim \text{Beta}(\alpha_{\text{TPB}}, \alpha_{\text{TPE}}) \qquad \text{draw rule type probabilities}$$

$$\phi_z^{(B)} \mid z, \boldsymbol{\beta} \sim \text{DP}(\alpha_{\text{BR}}, \boldsymbol{\beta}\boldsymbol{\beta}^{\text{T}}) \qquad \text{draw branching rule probabilities}$$

$$\phi_z^{(E)} \mid z, \boldsymbol{\gamma} \sim \text{DP}(\alpha_{\text{EM}}, \boldsymbol{\gamma}) \qquad \text{draw terminal production rule probabilities}$$

For each data string $j$,

$$r_j \mid \phi^{(R)} \sim \text{Categorical}\left(\phi^{(R)}\right) \qquad \text{choose root non-terminal}$$

For each node $i$ labeled with non-terminal $z_i$,

$$t_i \mid z_i, \phi_{z_i}^{(T)} \sim \text{Bernoulli}\left(\phi_{z_i}^{(T)}\right) \qquad \text{choose child type (non-terminals or terminal)}$$

$$\left(z_{c_1(i)}, z_{c_2(i)}\right) \mid z_i, \phi_{z_i}^{(B)}, t_i = 0 \sim \text{Categorical}\left(\phi_{z_i}^{(B)}\right) \qquad \text{generate child non-terminals}$$

$$s_i \mid z_i, \phi_{z_i}^{(E)}, t_i = 1 \sim \text{Categorical}\left(\phi_{z_i}^{(E)}\right) \qquad \text{generate child terminal}$$

(b)

$$\boldsymbol{\omega} \sim \text{GEM}(\alpha_{\text{State}}) \qquad \text{draw top-level state weights}$$

$$\phi^{(R)} \mid \boldsymbol{\omega} \sim \text{DP}(\alpha_{\text{Init}}, \boldsymbol{\omega}) \qquad \text{draw probabilities of initial states}$$

For each state $s$,

$$\phi_s^{(H)} \mid s \sim \text{Beta}(\alpha_{\text{Halt}}, \alpha_{\text{Move}}) \qquad \text{draw probability of halt at state } s$$

$$\phi_s^{(S)} \mid s, \boldsymbol{\omega} \sim \text{DP}(\alpha_{\text{TR}}, \boldsymbol{\omega}) \qquad \text{draw probabilities of transitions from state } s$$

For each data string $j$,

$$r_j \mid \phi^{(R)} \sim \text{Categorical}\left(\phi^{(R)}\right) \qquad \text{choose initial state}$$

For each time step $i$, given that the current state is $s_i$,

$$t_i \mid s_i, \phi_{s_i}^{(H)} \sim \text{Bernoulli}\left(\phi_{s_i}^{(H)}\right) \qquad \text{choose halt } (t_i = 0) \text{ or move-on } (t_i = 1)$$

$$s_{i+1} \mid s_i, \phi_{s_i}^{(S)}, t_i = 1 \sim \text{Categorical}\left(\phi_{s_i}^{(S)}\right) \qquad \text{choose next state}$$

(c)

for each terminal/state $s$,

$$\Sigma_s \mid s \sim \mathcal{W}^{-1}(\Psi_0, \nu_0) \qquad \text{draw covariance matrix parameters of the Gaussian emission}$$

$$\boldsymbol{\mu}_s \mid s, \Sigma_s \sim \mathcal{N}\left(\mathbf{m}_0, \frac{\Sigma_s}{k_0}\right) \qquad \text{draw mean parameters of the Gaussian emission}$$

For each node $i$ labeled with terminal/state $s_i$

$$\mathbf{x}_i \mid s_i, \mu_{s_i}, \Sigma_{s_i} \sim \mathcal{N}(\mu_{s_i}, \Sigma_{s_i}) \qquad \text{emit the MFCC vector}$$

**Figure 5.** HDP-PCFG with variable root and Gaussian emission conditioned on terminals ($a + c$), and HDP-HMM with variable initial state and Gaussian emission ($b + c$).

## 6.4. Posterior inference

Obtaining the true posterior distribution of the HDP-PCFG/HMM given data is computationally intractable. Accordingly, we approximated the posterior by *variational inference* [24,25,51,52]. Variational inference approximates the intractable posterior distribution $p(\theta|\mathbf{x})$, where $\theta$ bundles the latent variables of the PCFG/HMM in figure 5, by another distribution $q(\theta)$ that belongs to a class of

tractable distributions. We adopted the *mean-field method* and assumed the independence of the latent variables in $q(\theta)$, with $q(\theta)$ equal to the product of $q(\beta)$, $q(\gamma)$, $q(\phi^{(R)})$, $q(\phi_z^{(T)})$, $q(\phi_z^{(B)})$, $q(\phi_z^{(E)})$, $q(\tau)$ ($\tau := (\mathbf{r}, \mathbf{t}, \mathbf{z}, \mathbf{s})$) and $q(\mu_s, \Sigma_s)$ for the HDP-PCFG, with a similar factorization assumed for the HDP-HMM. The individual factor distributions were defined as follows. For all types of rule probabilities $\phi$, $q(\phi)$ was a Dirichlet distribution. $q(\tau)$ for parses $\tau$ was a multinomial distribution. $q(\mu_s, \Sigma_s)$ was a Gaussian-Inverse-Wishart distribution. Finally, $q(\beta)$ and $q(\gamma)$ were degenerate distributions such that they upper-bounded the possible number of (non-)terminals by positive integers $K_\beta$ and $K_\gamma$ respectively: $q(\beta_z > 0) = 0$ for $\forall z > K_\beta$ and $q(\gamma_s > 0) = 0$ for $\forall s > K_\gamma$. We set $K_\beta = 40$ and $K_\gamma = 100$, and similarly upper-bounded the possible number of HDP-HMM states by $K_\omega = 100$. Only 35 non-terminals, 64 terminals and 58 states had expected frequencies greater than 0.05 for the training data, and so we considered the truncation levels to be sufficiently large. Each variational factor distribution was updated iteratively so that the updates minimized the Kullback–Leibler (KL) divergence of the variational distribution $q(\theta)$ to the true posterior $p(\theta|\mathbf{x})$ (*coordinate ascent algorithm*). See [24,25] for the details on updates of the factor distributions for the HDP-PCFG/HMM latent variables (figure 5a,b), and [51,52] for those related to the Gaussian emission (figure 5c). See also [53] for updates of the degenerate factor distributions $q(\beta)$, $q(\gamma)$ and $q(\omega)$. The coordinate ascent algorithm only leads to a local minimum of the KL divergence, and accordingly, we tried 500 different sets of initial values for the parameters of $q$ and reported the best run among them. Each run terminated either when the improvement in the KL divergence was smaller than 0.1 or when the number of maximum iterations (=300) was achieved.

The statistical values reported in the Results section were estimated as follows based on the variational approximation. The expected count of a parse $\mathbf{z}_\iota$ of a training/test data string $\mathbf{x}_\iota$ was proportional to the exponential of its expected log probability, $\exp(E_{q(\theta_{\mathrm{PCFG}})}[\log p(\mathbf{z}_\iota \mid \mathbf{x}_\iota, \theta_{\mathrm{PCFG}})])$, where $\theta_{\mathrm{PCFG}}$ bundles the latent variables of the HDP-PCFG in figure 5a,c [25]. The Bayes-optimal parses are those maximize this expected log probability. While there are an intractable number of parse types, computation of the expected counts of the left-branching, right-branching and Bayes-optimal parses is tractable: we computed the exponential of their expected log probability and normalized it by the total, $\sum_{\mathbf{z}_\iota} \exp(E_{q(\theta_{\mathrm{PCFG}})}[\log p(\mathbf{z}_\iota \mid \mathbf{x}_\iota, \theta_{\mathrm{PCFG}})])$, which can be computed by dynamic programming (we adopted Earley's algorithm in particular [54,55]). The expected counts of the non-regular parses are just the total counts minus the expected counts of the left- and right-branching parses. The posterior predictive probability density of the test data under PCFG/HMM was given by $\log E_{q(\theta_{\mathrm{PCFG}})}[p(\mathbf{z}_\iota \mid \mathbf{x}_\iota, \theta_{\mathrm{PCFG}})]$ [56,57]. Finally, the expected number of production rules (root-labelling, branching, and terminal production rules for the PCFG; initial and inter-state transitions for the HMM) used for the training data were estimated by applying $\mathbb{1}[\cdot \geq 1]$ and tanh to the expected token frequency of the rules. The expected frequency of a rule $A \rightarrow \lambda$ was given by the parameters of its Dirichlet variational distribution $q(\phi_{A \rightarrow \lambda})$ minus its base counts $\alpha_{Em}\gamma_\lambda$ (for terminal production rules), $\alpha_{Br}\beta_B\beta_C$ (for branching rules such that $\lambda = (B, C)$), or $\alpha\beta_\lambda$ (for other rules).

Data accessibility. The data used in this paper are available at Dryad Digital Repository [58]. The code for the Preprocessing is available at https://github.com/hkoda/gibbon_peak_mfcc, and that for the Posterior Inference is available at https://bitbucket.org/tkc-morita/vi_hdphmm_hdppcfg.

Authors' contributions. T.M. participated in the data collection and performed the grammatical analyses of the data. H.K. participated in the data collection and performed the acoustic preprocessing of the data.

Competing interests. We have no competing interests.

Funding. This work is supported by the JSPS/MEXT KAKENHI(#18H03503 and #4903(Evolinguistic), JP17H06380) and JST CREST#17941861 (#JPMJCR17A4).

Acknowledgements. We thank Shigeru Miyagawa for discussions in the early stage of this study; Norihiko Maeda, Takumi Kunieda, Hiroshi Okuno, Kazuhiro Nakadai and the HARK project for their support in the recording process; the KUPRI Center for Human Evolution Modelling Research for their daily care of the gibbons.

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
