## [Reviewer comments · Royal Society Open Science]

Review History

RSOS-190139.R0 (Original submission)

Review form: Reviewer 1

Is the manuscript scientifically sound in its present form?

Yes

Are the interpretations and conclusions justified by the results?

Yes

Is the language acceptable?

Yes

Is it clear how to access all supporting data?

Yes

Do you have any ethical concerns with this paper?

No

Have you any concerns about statistical analyses in this paper?

No

Recommendation?

Accept with minor revision (please list in comments)

Comments to the Author(s)

The authors show animal song (to use their adopted terminology) can be analyzed as superregular (as is the case with human language). This being the case, it cannot be rejected that song and human language are governed by the same kind of grammar type.

As far as I can tell, both the analyses of the data and the procedures for acquiring it are sound and described to great detail. Relevant code is made readily available, and details on future availability of relevant data are given.

I believe the paper warrants publication, and that it would provide a welcome contribution to fields concerned with complexity and evolution of human and animal cognition. Below I'll provide some comments that, if addressed, would add to the value of the paper, in my view.

1. I call attention to a case of superregular analysis on a non-human species, a PCFG analysis of humpback whales: https://officerloud.files.wordpress.com/2018/12/9563B_LingDiss-1-1.pdf

While the authors of the paper under review don't have to agree with the exact methods or conclusions of humpback whale analyses, perhaps it would be worth mentioning it, since it is an attempt in the same direction. (for the sake of transparency, I am not the author of the paper).

2. As the text and the title itself make abundantly clear, the goal of the paper is to show that both human language and animal song *can* be subject to suprarregular analyses, and there are advantages for doing so (namely, more compact analyses). Perhaps the authors consciously made the decision of keeping it at that instead of risking "opening a can of worms" and derailing the merit of the paper with more ambitious claims. Still: there are important evolutionary claims here that could be at least mentioned. If the centrality of superregularity as the thing that makes human language unique is put to question, the burden of proof is put on Berwick & Chomsky's (B&C) (and colleagues) line of reasoning. It either: I) forces a different description that is indeed only applicable to human language, so as to retain its computational uniqueness, which then must be explained evolutionarily or II) strengthens the view that human language is made up of properties that are found in other species and domains, which would lend support to the idea that a comparative biology approach is the one best suited as a way of explaining how human language came about. There are some passages that make reference to the work of B&C, but emphasizing the evolutionary implications would make the claim stronger and indeed, more wide-ranging.

I believe this evolutionary challenge to the B&C view holds even if we take the extremely rare and convoluted " $m \geq 3$ " cases in human language into account: it could be that these are rare for, say, working memory reasons (thus not for "grammatical" reasons) and that animal song has limitations of the same kind that we don't know about. Given that, as the authors show, we can successfully employ superregular analysis of song, and even have good reason to do so, it's an empirical question whether the human vs animal systems are indeed different in that regard, and a comparative approach should be the most fruitful approach.

3. I advise against using pie charts for figure 2. Since each slice has to be labelled with the correct percentages, defeating the purpose of a graph in the first place, perhaps the pie chart form is not informative aesthetically. Maybe a bar chart, such as the ones used in Fig 3, would be a better representation of the information.

Review form: Reviewer 2

Is the manuscript scientifically sound in its present form?

No

Are the interpretations and conclusions justified by the results?

No

Is the language acceptable?

Yes

Is it clear how to access all supporting data?

Yes

Do you have any ethical concerns with this paper?

No

Have you any concerns about statistical analyses in this paper?

No

Recommendation?

Major revision is needed (please make suggestions in comments)

Comments to the Author(s)

Major comments:

This paper claims that the previous argument that animal song is regular lacks empirical evidence, but I don't understand what "empirical" means here. Then, does the previous literature provide no evidence at all to support the argument? Please elaborate on or maybe review "original" studies on animal song cited in the literature.

The relationship between HDP prior and grammar compactness is extremely crucial for the conclusion of this paper. Please elaborate on this at an intuitive, not mathematical, level.

It seems that PCFG and HMM are induced without any explicit supervision, so there is no guarantee that Viterbi parses are "correct" parses of strings. Please explain how rules were induced and parses were categorized into regular and non-regular (the grammar induction procedure is not explained in detail in the first place).

The method to compute log ratios sounds cherry-picking. The results may not change, but the data without non-regular parses should be included with probabilities of non-regular parses being zero.

The table like Perfors et al. (2011) should be shown for readers to acknowledge the results.

Importantly, if the posteriors are almost the same between two models and the prior was larger with PCFG than HMM, then is the likelihood larger with HMM than PCFG, meaning that HMM is more explanatory?

The stance of this paper is unclear regarding "a methodological question whether animal song should be analyzed by explanatory power alone or by Bayesian posterior inference". Crucially for the purpose here, grammar compactness doesn't necessarily mean that generative capacity of gibbon "grammar" is superregular. In other words, methodological simplicity should be distinguished from ontological and biological claims.

Even if the conclusion of this paper is granted, then what distinguishes human language from animal song? Please discuss broader implications of this paper.

The amount of overlap among three gibbon songs should be empirically justified. Otherwise, the combination of three songs may increase computational complexity to the context-free range in an unpredictable way.

Minor comments:

p.1: Why the term "superregular", not "context-free", given that PCFG is clearly context-free by definition.

p.2: Arguments from corpus and processing don't have to be cited to cast doubt on the idealization; infinity cannot be just observed.

p.2: I don't understand the sentence "A better empirical argument for the superregularity of human language comes from probabilistic analysis". Better in what respect?

p.2: What is "constant" in the equation? And with this constant, the equation shouldn't be exact in the technical sense.

p.3: What does "active" as in "active rules" mean exactly?

p.3: I don't understand the sentence "Note that regular grammars are a subclass of PCFG and, thus, are included in our hypothesis space". The inclusion of regular languages by context-free languages is not an issue here.

p.3: The phrase "...which is the standard evaluation metric for language models in the field of natural language processing (NLP)" seems unnecessary, simple because this paper is clearly not a NLP paper.

p.3: The length of at least 3, not 4, is minimally required to create non-regular parses.

p.3: Regarding the sentence "the non-regular parses were significantly more probable than the regular parses", is the "significance" statistical? In general, no statistics are reported in this paper.

p.4: The amount of training and test data looks unbalanced. Any reason(s)?

p.4: Despite the fact that no left-branching parses were Viterbi parses, the log ratio for left-branching was smaller than the one for right-branching. Why?

p.5: See Berwick (2015) "Mind the Gap" for an argument on grammar compactness.

p.6: Why only first channels of the recordings?

p.7: How does variable root PCFG make sure that the grammar is consistent in the technical sense?

Review form: Reviewer 3

Is the manuscript scientifically sound in its present form?

Yes

Are the interpretations and conclusions justified by the results?

Yes

Is the language acceptable?

Yes

Is it clear how to access all supporting data?

Yes

Do you have any ethical concerns with this paper?

No

Have you any concerns about statistical analyses in this paper?

Yes

Recommendation?

Major revision is needed (please make suggestions in comments)

Comments to the Author(s)

Please see attached report (Appendix A).

Decision letter (RSOS-190139.R0)

12-Apr-2019

Dear Dr Morita,

The editors assigned to your paper ("Superregular grammars do not provide additional explanatory power but allow for a compact analysis of animal song") have now received comments from reviewers. We would like you to revise your paper in accordance with the referee and Associate Editor suggestions which can be found below (not including confidential reports to the Editor). Please note this decision does not guarantee eventual acceptance.

Please submit a copy of your revised paper before 05-May-2019. Please note that the revision deadline will expire at 00.00am on this date. If we do not hear from you within this time then it

will be assumed that the paper has been withdrawn. In exceptional circumstances, extensions may be possible if agreed with the Editorial Office in advance. We do not allow multiple rounds of revision so we urge you to make every effort to fully address all of the comments at this stage. If deemed necessary by the Editors, your manuscript will be sent back to one or more of the original reviewers for assessment. If the original reviewers are not available, we may invite new reviewers.

- Data accessibility

If you wish to submit your supporting data or code to Dryad (<http://datadryad.org/>), or modify your current submission to dryad, please use the following link:
<http://datadryad.org/submit?journalID=RSOS&manu=RSOS-190139>

- Competing interests

- Authors' contributions

- Acknowledgements

- Funding statement

Kind regards,

Andrew Dunn

on behalf of Dr Claudia Wascher (Associate Editor) and Kevin Padian (Subject Editor)

Associate Editor's comments (Dr Claudia Wascher):

Associate Editor: 1

Comments to the Author:

The presented manuscript aims at investigating computational complexity in animal song and provides a case study to demonstrate that a superregular analysis can be effectively used with non-human data. As such, the reviewer and I agree that the present paper is valuable. However, the reviewers do raise concerns, which need to be addressed prior to publication. More specifically, concerns regarding the segmentation of calls and discussion of the broader and specifically evolutionary implications of the paper.

Comments to Author:

Reviewers' Comments to Author:

Reviewer: 1

Comments to the Author(s)

The authors show animal song (to use their adopted terminology) can be analyzed as superregular (as is the case with human language). This being the case, it cannot be rejected that song and human language are governed by the same kind of grammar type.

As far as I can tell, both the analyses of the data and the procedures for acquiring it are sound and described to great detail. Relevant code is made readily available, and details on future availability of relevant data are given.

I believe the paper warrants publication, and that it would provide a welcome contribution to fields concerned with complexity and evolution of human and animal cognition. Below I'll provide some comments that, if addressed, would add to the value of the paper, in my view.

1. I call attention to a case of superregular analysis on a non-human species, a PCFG analysis of humpback whales: https://officerloud.files.wordpress.com/2018/12/9563B_LingDiss-1-1.pdf

While the authors of the paper under review don't have to agree with the exact methods or conclusions of humpback whale analyses, perhaps it would be worth mentioning it, since it is an attempt in the same direction. (for the sake of transparency, I am not the author of the paper).

2. As the text and the title itself make abundantly clear, the goal of the paper is to show that both human language and animal song *can* be subject to suprarregular analyses, and there are advantages for doing so (namely, more compact analyses). Perhaps the authors consciously made the decision of keeping it at that instead of risking "opening a can of worms" and derailing the merit of the paper with more ambitious claims. Still: there are important evolutionary claims here that could be at least mentioned. If the centrality of superregularity as the thing that makes human language unique is put to question, the burden of proof is put on Berwick & Chomsky's (B&C) (and colleagues) line of reasoning. It either: I) forces a different description that is indeed only applicable to human language, so as to retain its computational uniqueness, which then must be explained evolutionarily or II) strengthens the view that human language is made up of properties that are found in other species and domains, which would lend support to the idea that a comparative biology approach is the one best suited as a way of explaining how human language came about. There are some passages that make reference to the work of B&C, but emphasizing the evolutionary implications would make the claim stronger and indeed, more wide-ranging.

I believe this evolutionary challenge to the B&C view holds even if we take the extremely rare and convoluted " $m \geq 3$ " cases in human language into account: it could be that these are rare for, say, working memory reasons (thus not for "grammatical" reasons) and that animal song has limitations of the same kind that we don't know about. Given that, as the authors show, we can successfully employ superregular analysis of song, and even have good reason to do so, it's an empirical question whether the human vs animal systems are indeed different in that regard, and a comparative approach should be the most fruitful approach.

3. I advise against using pie charts for figure 2. Since each slice has to be labelled with the correct percentages, defeating the purpose of a graph in the first place, perhaps the pie chart form is not informative aesthetically. Maybe a bar chart, such as the ones used in Fig 3, would be a better representation of the information.

Reviewer: 2

Comments to the Author(s)

Major comments:

This paper claims that the previous argument that animal song is regular lacks empirical evidence, but I don't understand what "empirical" means here. Then, does the previous literature

provide no evidence at all to support the argument? Please elaborate on or maybe review "original" studies on animal song cited in the literature.

The relationship between HDP prior and grammar compactness is extremely crucial for the conclusion of this paper. Please elaborate on this at an intuitive, not mathematical, level.

It seems that PCFG and HMM are induced without any explicit supervision, so there is no guarantee that Viterbi parses are "correct" parses of strings. Please explain how rules were induced and parses were categorized into regular and non-regular (the grammar induction procedure is not explained in detail in the first place).

The method to compute log ratios sounds cherry-picking. The results may not change, but the data without non-regular parses should be included with probabilities of non-regular parses being zero.

The table like Perfors et al. (2011) should be shown for readers to acknowledge the results. Importantly, if the posteriors are almost the same between two models and the prior was larger with PCFG than HMM, then is the likelihood larger with HMM than PCFG, meaning that HMM is more explanatory?

The stance of this paper is unclear regarding "a methodological question whether animal song should be analyzed by explanatory power alone or by Bayesian posterior inference". Crucially for the purpose here, grammar compactness doesn't necessarily mean that generative capacity of gibbon "grammar" is superregular. In other words, methodological simplicity should be distinguished from ontological and biological claims.

Even if the conclusion of this paper is granted, then what distinguishes human language from animal song? Please discuss broader implications of this paper.

The amount of overlap among three gibbon songs should be empirically justified. Otherwise, the combination of three songs may increase computational complexity to the context-free range in an unpredictable way.

Minor comments:

p.1: Why the term "superregular", not "context-free", given that PCFG is clearly context-free by definition.

p.2: Arguments from corpus and processing don't have to be cited to cast doubt on the idealization; infinity cannot be just observed.

p.2: I don't understand the sentence "A better empirical argument for the superregularity of human language comes from probabilistic analysis". Better in what respect?

p.2: What is "constant" in the equation? And with this constant, the equation shouldn't be exact in the technical sense.

p.3: What does "active" as in "active rules" mean exactly?

p.3: I don't understand the sentence "Note that regular grammars are a subclass of PCFG and, thus, are included in our hypothesis space". The inclusion of regular languages by context-free languages is not an issue here.

p.3: The phrase "...which is the standard evaluation metric for language models in the field of natural language processing (NLP)" seems unnecessary, simple because this paper is clearly not a NLP paper.

p.3: The length of at least 3, not 4, is minimally required to create non-regular parses.

p.3: Regarding the sentence "the non-regular parses were significantly more probable than the regular parses", is the "significance" statistical? In general, no statistics are reported in this paper.

p.4: The amount of training and test data looks unbalanced. Any reason(s)?

p.4: Despite the fact that no left-branching parses were Viterbi parses, the log ratio for left-branching was smaller than the one for right-branching. Why?

p.5: See Berwick (2015) "Mind the Gap" for an argument on grammar compactness.

p.6: Why only first channels of the recordings?

p.7: How does variable root PCFG make sure that the grammar is consistent in the technical sense?

Reviewer: 3

Comments to the Author(s)
Please see attached report.

Author's Response to Decision Letter for (RSOS-190139.R0)

See Appendix B.

RSOS-190139.R1 (Revision)

Review form: Reviewer 1

Is the manuscript scientifically sound in its present form?

Yes

Are the interpretations and conclusions justified by the results?

Yes

Is the language acceptable?

Yes

Is it clear how to access all supporting data?

Yes

Do you have any ethical concerns with this paper?

No

Have you any concerns about statistical analyses in this paper?

No

Recommendation?

Accept as is

Comments to the Author(s)

Thank for addressing all comments.

I understand the choice not to include discussion of the work on humpback whales (Kennedy 2018), given its uncertain peer-review, publication and availability status. And since this very exchange will be made public, the discussion will be available, as the authors point out. I therefore agree with the authors that it is not necessary to move the discussion to the main text of the paper. It is clear that the authors engaged with the material in a detailed manner.

My other comments were also addressed adequately (namely, brief discussion on evolutionary and comparative matters, which the authors included without running the risk of derailing the main point of the paper, and some advice on visualization of figure 2.)

Extensive comments by reviewer #2 and #3, more technical in nature than the ones I offered, were also diligently addressed.

I think this will be an important paper and support its publication.

Review form: Reviewer 3

Is the manuscript scientifically sound in its present form?

Yes

Are the interpretations and conclusions justified by the results?

Yes

Is the language acceptable?

Yes

Is it clear how to access all supporting data?

Yes

Do you have any ethical concerns with this paper?

No

Have you any concerns about statistical analyses in this paper?

No

Recommendation?

Accept with minor revision (please list in comments)

Comments to the Author(s)

Dear Authors,

Overall, I think you did a nice job addressing most of my concerns. I do not notice some minor errors, such as the omission of Fig 4 which displays as blank (at least with the Chrome and Acrobat PDF viewers) and it would be nice to see what you downsampled to, but that can be inferred.

The only concern of any consequence that is still bothering me is how much the use of acoustic vectors as opposed to symbols influences the analysis. I like the justifications that you have added, but I can't help thinking that it might be more convincing if the same analysis was done on the human infant data base to which you compare. I did not raise this concern the first time I reviewed this as it was not until I read it this time to recognize that this was at the crux of what bothered me about the analysis. I don't know that this analysis is absolutely necessary, I think that you raise interesting ideas and the work stands on its own, but it might be a stronger argument if you could make the apples and apples comparison in future work.

Decision letter (RSOS-190139.R1)

13-Jun-2019

Dear Dr Morita:

On behalf of the Editors, I am pleased to inform you that your Manuscript RSOS-190139.R1 entitled "Superregular grammars do not provide additional explanatory power but allow for a compact analysis of animal song" has been accepted for publication in Royal Society Open Science subject to minor revision in accordance with the referee suggestions. Please find the referees' comments at the end of this email.

The reviewers and Subject Editor have recommended publication, but also suggest some minor revisions to your manuscript. Therefore, I invite you to respond to the comments and revise your manuscript.

- Ethics statement

- Data accessibility

If you wish to submit your supporting data or code to Dryad (<http://datadryad.org/>), or modify your current submission to dryad, please use the following link:
<http://datadryad.org/submit?journalID=RSOS&manu=RSOS-190139.R1>

- **Competing interests**

- **Authors' contributions**

- **Acknowledgements**

- **Funding statement**

Because the schedule for publication is very tight, it is a condition of publication that you submit the revised version of your manuscript before 22-Jun-2019. Please note that the revision deadline will expire at 00.00am on this date. If you do not think you will be able to meet this date please let me know immediately.

When submitting your revised manuscript, you will be able to respond to the comments made by the referees and upload a file "Response to Referees" in "Section 6 - File Upload". You can use this to document any changes you make to the original manuscript. In order to expedite the

processing of the revised manuscript, please be as specific as possible in your response to the referees.

on behalf of Dr Claudia Wascher (Associate Editor) and Kevin Padian (Subject Editor)
openscience@royalsociety.org

Associate Editor Comments to Author (Dr Claudia Wascher):

The authors provide a thorough revision of their manuscript. One reviewer provides a further suggestions, which I think would be useful for the authors to consider prior to publication.

Reviewer comments to Author:
Reviewer: 1

Comments to the Author(s)
Thank for addressing all comments.

I understand the choice not to include discussion of the work on humpback whales (Kennedy 2018), given its uncertain peer-review, publication and availability status. And since this very exchange will be made public, the discussion will be available, as the authors point out. I therefore agree with the authors that it is not necessary to move the discussion to the main text of the paper. It is clear that the authors engaged with the material in a detailed manner.

My other comments were also addressed adequately (namely, brief discussion on evolutionary and comparative matters, which the authors included without running the risk of derailing the main point of the paper, and some advice on visualization of figure 2.)

Extensive comments by reviewer #2 and #3, more technical in nature than the ones I offered, were also diligently addressed.

I think this will be an important paper and support its publication.

Reviewer: 3

Comments to the Author(s)

Dear Authors,

Overall, I think you did a nice job addressing most of my concerns. I do not notice some minor errors, such as the omission of Fig 4 which displays as blank (at least with the Chrome and Acrobat PDF viewers) and it would be nice to see what you downsampled to, but that can be inferred.

The only concern of any consequence that is still bothering me is how much the use of acoustic vectors as opposed to symbols influences the analysis. I like the justifications that you have added, but I can't help thinking that it might be more convincing if the same analysis was done on the human infant data base to which you compare. I did not raise this concern the first time I reviewed this as it was not until I read it this time to recognize that this was at the crux of what bothered me about the analysis. I don't know that this analysis is absolutely necessary, I think that you raise interesting ideas and the work stands on its own, but it might be a stronger argument if you could make the apples and apples comparison in future work.

Author's Response to Decision Letter for (RSOS-190139.R1)

See Appendix C.

Decision letter (RSOS-190139.R2)

14-Jun-2019

Dear Dr Morita,

I am pleased to inform you that your manuscript entitled "Superregular grammars do not

provide additional explanatory power but allow for a compact analysis of animal song" is now accepted for publication in Royal Society Open Science.

on behalf of Dr Claudia Wascher (Associate Editor) and Kevin Padian (Subject Editor)
openscience@royalsociety.org

Follow Royal Society Publishing on Twitter: [@RSocPublishing](https://twitter.com/RSocPublishing)
Follow Royal Society Publishing on Facebook:
<https://www.facebook.com/RoyalSocietyPublishing.FanPage/>
Read Royal Society Publishing's blog: <https://blogs.royalsociety.org/publishing/>

Appendix A

Dear Drs. Morita and Koda,

Thank you for sharing your work addressing the complexity of grammars to represent non-human song. I found your case study with gibbons to be well written, interesting, and thought provoking. To briefly summarize your work, you build on the methods of Perfors et al. (2011) who hypothesized that the complexity of a grammar (or a generating principle in their case) could be represented by the prior probability of selecting a specific grammar from a family of grammars. The actual process of estimating this probability is tricky and your paper uses variational methods to estimate this. I am not a linguist and am not in a good position to judge how effective this is, but it seems clear that other scholars have employed this method and will defer to other reviewers with respect to the appropriateness of whether or not this specific prior estimate is merited.

You compared the following types of grammars: probabilistic context free grammars with hierarchical Dirichlet priors (with multiple start symbols) which represented the superregular case and hidden Markov models with the same priors, representing the regular case. In both cases, rather than using discrete symbols for song notes, your observations were Mel-filtered cepstral coefficients (MFCC) spanning 100 ms song regions. The 100 ms regions were first identified by a standard energy ratio detector and further refined by searching for peaks in summed envelopes of a signal filterbank output.

While I have heard gibbon calls before, I am not intimately familiar with them (I am a bioacoustian who works on other taxa). From my memory and a quick review of some of the spectrograms in Clarke *et al.* (2006)'s well-cited paper on syntax and meaning of pileated gibbons (*Hylobates pileatus*; not *H. agilis* & *H. albibarbis* as used in your study), 100 ms seems reasonably appropriate to me although it is clear that some longer calls will be broken into multiple 100 ms segments. I think it would be appropriate to include a figure that provides examples of the types of segmentations resulting from this type of analysis. The peak selection could also use further clarification. What does it mean to be "greater than 1.0 of the minimum in a 300 ms search window" (line 52). It is unclear what the units are (counts, dB re. 20 μ Pa?), making this comparison meaningless.

The decision to use MFCC and 100 ms frames has several repercussions. On the positive side, it may address the graded vs. discrete call conundrum that we face with non-human calls. On the negative side, long calls will be broken down into low-level acoustic units. When one considers a longer call such as the following:

which is roughly sketched from the aforementioned Clarke *et al.* paper with a duration of 1.5 s, there will be 100 ms segments covering things that might be likely to be a single structure, even if one entertains the hypothesis that there may be meaningful subcomponents to this call. This is a bit of a

cause of concern to me. I think that this will be likely to introduce some very deterministic structure into the grammars that are generated. It also raises questions (to me at least) about what kind of information was captured and how meaningful it might be.

The conclusion was that while regular grammars gave roughly equivalent modeling capabilities (slightly better) than superregular grammars, the superregular grammars had higher prior probabilities suggesting that they are a more efficient representation. While I do have some concerns, as I stated in the opening of this review, I find your work to be intriguing and would like to see refinements in the methods as I think your work is interesting and has potential.

Thank you for the read.

Specific comments:

p1 | 41 & 52 – cite

p3 | 43 – I believe the parse induction experiment is carried out only on the PCFG grammar and the separation is just looking at non-regular vs right-branching rules. If correct, please state, otherwise clarify

p 4 | 50 - Where does the 77 from 52-77 come from?

p 4 | 56 – The conclusion seems a bit overstated to me.

Fig 4 – It took me a while to figure out this bar plot. Perhaps change styles (point +/- 95% CI) and make the axis not span 0 to -40?

p 6 | 40 – A 20 ms window on a signal with $F_s=16$ kHz is 320 samples. This is not a power of 2, so you probably did not use the fast Fourier transform. I suggest changing to discrete Fourier transform (DFT).

Authors' Responses to the Comments from the Reviewers on ID RSOS-190139, "Superregular grammars do not provide additional explanatory power but allow for a compact analysis of animal song"

Takashi Morita

Hiroki Koda

Dear Dr. Claudia Wascher, our handling editor of Royal Society Open Science,

Thank you so much for giving us a chance to improve our manuscript. All the comments from the reviewers were insightful, constructive, and really helpful to revise our manuscript. We made every possible effort to meet the requests from the reviewers. The rest of this letter first summarizes what we revised, and then responds to the individual comments from the reviewers.

Summary of the Major Revisions

The following major revisions were made in the manuscript.

- We now discuss *evolutionary implications of the paper* in the section "5. General Discussion and Conclusion". We adopted Reviewer #1's suggestion and argue that comparative studies could be more meaningful than theoretical linguists have claimed once we analyze animal data with PCFG and other superregular models.
- The subsection "(b) Preprocessing" under "6. Materials & Methods" now explains the motivation behind the segmentation and acoustic feature extraction of gibbon song in more detail.
- Given the complexity of our Materials & Methods, we added a *new section "3. Overview of the Materials & Methods"*, where we *outline our grammatical analysis without technical details*. We hope it helps readers to interpret our results more easily.
- Following reviewers' suggestions, the first result section, "4(a) Posterior Inference of PCFG", has new presentation of the same result.

Responses to the Associate Editor's comments:

... the reviewers do raise concerns, which need to be addressed prior to publication. ... concerns regarding the segmentation of calls

The subsection "(b) Preprocessing" under "6. Materials & Methods" now explains the motivation behind the segmentation and acoustic feature extraction of gibbon song in more detail.

and discussion of the broader and specifically evolutionary implications of the paper.

We now discuss evolutionary implications of the paper in the section "5. General Discussion and Conclusion".

Responses to Reviewer 1

Thank you for your constructive comments. Your suggestion about evolutionary implications of our paper was really helpful in particular. Please find our responses to your comments below.

1. I call attention to a case of superregular analysis on a non-human species, a PCFG analysis of humpback whales: https://officerloud.files.wordpress.com/2018/12/9563B_LingDiss-1-1.pdf While the authors of the paper under review don't have to agree with the exact methods or conclusions of humpback whale analyses, perhaps it would be worth mentioning it, since it is an attempt in the same direction. (for the sake of transparency, I am not the author of the paper).

Thank you so much for letting us know this work (Kennedy, 2018, reference info provided below for readers of our work). While the paper is an valuable attempt of a superregular analysis of animal data, however, we are a bit concerned that it is an unpublished undergraduate thesis. As you probably know, not everyone considers it appropriate to cite such a work in a journal article. And we're particularly concerned that the thesis referred to by Reviewer #1 is not uploaded to a repository hosted by Cambridge University or other reliable services like arXiv, but instead to the author's personal blog (<https://officerloud.com/>), and it is unclear to us if it would remain available over years.

Accordingly, let us limit discussion on the thesis to this revision letter. Royal Society Open Science publishes it along with the main manuscript, so readers would still be able to find our discussion. (We will move the discussion to the main manuscript, however, if you recommend it.)

The thesis proposes a CFG built based on three strings of manually transcribed sound units (categorized as Phrase C1) produced by a single humpback whale. The author reports that the proposed CFG can parse all the three instances of C1 without "overgeneration": The CFG does not generate "unattested" substrings or parse any of 1000 random permutations of one of the three data strings. The author also states that an alternative regular analysis (a bigram model in particular) proposed by Hurford (2012) overgenerates "unattested" phrases.

We first have to point out that the data discussed in the thesis are extremely limited and the proposed grammar could actually be overfitted to the data. In other words, the CFG may fail to parse some instances of C1 when a much greater amount of the data are tested.

Nevertheless, the author's characterization of the CFG analysis is in accordance with what we discovered in our PCFG analysis of gibbon song: (P)CFG has a good balance between fit to data (likelihood) and compactness (prior). To clarify this point, let us first consider several possible regular analyses of humpback whales. As the author of the thesis points out, Hurford's finite-state automaton can generate many patterns that do not appear in the data (though this may turn out to be a good generalization when more data are analyzed). However, such overgeneration is not a general pattern applicable to *all* kinds of regular grammars: While the author only considers bigram models (which meets the first-order Markov property), finite state automata with a greater number of states (e.g., higher order ngram models) allow for the perfect fit to the training data. In an extreme case, we can consider what Perfors et al. (2011) call the FLAT grammar, which lists/memorizes all and only the observed strings and imposes a uniform distribution over the strings. The FLAT grammar is regular, but as demonstrated by Perfors et al. with human language data, it enables the greatest fit to the training data (i.e., likelihood). However, description of the FLAT grammar gets extremely long as the data size increases and a greater variety of string types are observed, and accordingly, the FLAT grammar (with non-zero likelihood) has a small prior probability. High order ngram models have similar problems, with exponential increase in the number of states of the automata. (Our HMM analysis also resulted in the same pattern: Its predictive performance was

as good as, or even slightly better than the PCFG, but the HMM needed to use a greater number of production rules to achieve it.) The CFG proposed in the thesis, on the other hand, can generate more than just the observed training data without increasing its description length. Hence, the CFG would be a balanced hypothesis with moderate fit to data (if not overfitted) and compactness.

Reference (Information for Readers):

Kennedy, Malcom (2018). Recursion in Humpback whale song (Unpublished Honours Thesis). Cambridge University, Cambridge, UK. Retrieved from: https://officerloud.files.wordpress.com/2018/12/9563B_LingDiss-1-1.pdf

2. As the text and the title itself make abundantly clear, the goal of the paper is to show that both human language and animal song *can* be subject to suprarregular analyses, and there are advantages for doing so (namely, more compact analyses). Perhaps the authors consciously made the decision of keeping it at that instead of risking "opening a can of worms" and derailing the merit of the paper with more ambitious claims. Still: there are important evolutionary claims here that could be at least mentioned. If the centrality of superregularity as the thing that makes human language unique is put to question, the burden of proof is put on Berwick & Chomsky's (B&C) (and colleagues) line of reasoning. It either: I) forces a different description that is indeed only applicable to human language, so as to retain its computational uniqueness, which then must be explained evolutionarily or II) strengthens the view that human language is made up of properties that are found in other species and domains, which would lend support to the idea that a comparative biology approach is the one best suited as a way of explaining how human language came about. There are some passages that make reference to the work of B&C, but emphasizing the evolutionary implications would make the claim stronger and indeed, more wide-ranging. I believe this evolutionary challenge to the B&C view holds even if we take the extremely rare and convoluted " $m \geq 3$ " cases in human language into account: it could be that these are rare for, say, working memory reasons (thus not for "grammatical" reasons) and that animal song has limitations of the same kind that we don't know about. Given that, as the authors show, we can successfully employ superregular analysis of song, and even have good reason to do so, it's an empirical question whether the human vs animal systems are indeed different in that regard, and a comparative approach should be the most fruitful approach.

We now discuss evolutionary implications of the paper in the section "5. General Discussion and Conclusion", following your line and emphasizing the importance of comparative studies. Thank you so much for your suggestion.

3. I advise against using pie charts for figure 2. Since each slice has to be labelled with the correct percentages, defeating the purpose of a graph in the first place, perhaps the pie chart form is not informative aesthetically. Maybe a bar chart, such as the ones used in Fig 3, would be a better representation of the information.

The pie charts were replaced with stacked bar plots in Figure 2, which now reports the expected counts of parses in an unnormalized way. (The information reported in the previous Figure 3 is also merged there.) We were not able to include the percentages in the figure, so the info is now in the text (in the section "4(a) Posterior Inference of PCFG").

Responses to Reviewer 2

Thank you first of all for your detailed and helpful comments. Please find our responses below.

Major comments:

This paper claims that the previous argument that animal song is regular lacks empirical evidence, but I don't understand what "empirical" means here. Then, does the previous literature provide no evidence at all to support the argument? Please elaborate on or maybe review "original" studies on animal song cited in the literature.

The Introduction section now briefly reviews previous studies on Bengalese finch song, which is the most popular example discussed in the literature. The models used in those studies are listed, and we point out that they are all regular and no superregular analysis has been performed on the species.

The relationship between HDP prior and grammar compactness is extremely crucial for the conclusion of this paper. Please elaborate on this at an intuitive, not mathematical, level.

Exactly. We now have a new section "3. Overview of the Materials & Methods", and we outline our grammatical analysis there without technical details.

It seems that PCFG and HMM are induced without any explicit supervision, so there is no guarantee that Viterbi parses are "correct" parses of strings. Please explain how rules were induced and parses were categorized into regular and non-regular (the grammar induction procedure is not explained in detail in the first place).

The new overview section clarifies that

1. Unlike human language, we have no reasonable CFG (a set of production rules) for gibbon song.
2. So the rule types are induced in the Bayesian manner as well as the rules' production probability.

As reported in the Materials & Methods section, the actual induction of PCFG rules was embedded in the variational approximation of the joint posterior. The current draft still omits the details of the variational inference (the coordinate ascent algorithm) because:

- Readers can find all the information in Liang et al. (2007, 2019) [18,19]. Detailed description would just duplicate their work and thus seem unnecessary.
- Our implementation of the algorithm is available in github.

However, we would report the details in Supporting Information etc. if you still think they are necessary.

You can find information related to the categorization of the regular vs. non-regular parses in the last paragraph of the subsection "6(d) Posterior Inference", which reports how we computed the expected counts of parses. In fact, the categorization of a given parse does not have any tricks: We can just check whether it is left/right-branching or not. Computation of the expected counts of left/right-branching parses is also tractable, and by subtracting them from the total, we can also compute the expected counts of non-regular parses.

The method to compute log ratios sounds cherry-picking. The results may not change, but the data without non-regular parses should be included with probabilities of non-regular parses being zero.

The parse analysis section, "4(a) Posterior Inference of PCFG", has new presentation of the same result. In particular, the new Figure 2 visualizes:

- The non-regular parses have greater expected counts than the regular parses even when we look at the whole data (no screening of the data any more, though the short data of length 1 or 2 are reported as a separate category since they are both left- and right-branching).
- The expected counts of non-regular Bayes-optimal parses is an order of magnitude greater than the chance level. (BTW, we decided not to use the term "Viterbi" because the term is based on the maximum likelihood analysis while our optimal parse is based on the posterior. We now use the term "Bayes-optimal" following Liang et al. (2019) [19].)

The table like Perfors et al. (2011) should be shown for readers to acknowledge the results. Importantly, if the posteriors are almost the same between two models and the prior was larger with PCFG than HMM, then is the likelihood larger with HMM than PCFG, meaning that HMM is more explanatory?

We changed the presentation in the section "2. Preliminaries" such that we clarify that

- the evaluation metric used by Perfors et al. (2011) is the posterior, and
- we can see what made the PCFG more probable than the regular grammars when we look at the components of the posterior: that is, the prior and the likelihood.

In the new section "3. Overview of the Materials & Methods", we also clarified that we also do the same reasoning: (i) We first show that superregular analyses are probable in the posterior (Section 4(a)); and (ii) then we analyze what made the superregular analyses probable (Section 4(b) and (c)).

The stance of this paper is unclear regarding "a methodological question whether animal song should be analyzed by explanatory power alone or by Bayesian posterior inference". Crucially for the purpose here, grammar compactness doesn't necessarily mean that generative capacity of gibbon "grammar" is superregular. In other words, methodological simplicity should be distinguished from ontological and biological claims.

The last paragraph of the General Discussion and Conclusion now emphasizes that the induced (posterior of) PCFG is just a computationally optimal hypothesis for the gibbon data at this point and is not guaranteed to match the actual system behind gibbon song.

In addition, we also emphasize that sticking to the PCFG analysis in this paper would cause the same problem as the previous assumption of regularity for animal data. We encourage future studies on animal song to consider more recent methods used in computational and theoretical linguistics, such as mildly-context sensitive grammars and advanced versions of PCFG (adaptor grammars).

Even if the conclusion of this paper is granted, then what distinguishes human language from animal song? Please discuss broader implications of this paper.

We still do not propose any specific differences between human language and animal song. However, we do discuss evolutionary implications of the paper now in the section "5. General Discussion and Conclusion", following the line suggested by Reviewer 1 (emphasizing the importance of comparative studies).

The amount of overlap among three gibbon songs should be empirically justified. Otherwise, the combination of three songs may increase computational complexity to the context-free range in an unpredictable way.

In the subsection "(a) Data" under "6. Materials and Methods", we now report that we actually attempted localization of the sound sources (i.e., singer identification) and separation of sounds from different sources, using

the HARK programs (<https://www.hark.jp/>). However, neither the localization nor separation was successful in our recording environment (also reported in the subsection), and for this technical reason, we were not able to assess the amount of overlap.

On the other hand, we point out that switching among multiple finite-state automata is equivalent to a single large finite-state automaton. Hence, data strings produced multiple individuals do not immediately lead to superregularity, though our approach is probabilistic and thus this formal language theoretic proof does not completely reject the possibility of superregularity due to overlap.

Minor comments:

p.1: Why the term "superregular", not "context-free", given that PCFG is clearly context-free by definition.

As you pointed out, the PCFG induced in our study is not guaranteed to match the actual system behind gibbon song. It remains possible that gibbon song is better analyzed by super-context-free grammars. The main argument in this paper is that we should not stipulate the regularity of animal song without empirical evidence, and similarly, we should not stick to PCFG analyses of animal song for comparative studies given that linguists have suggested insufficiency of (P)CFG for human language. These points are discussed in the last paragraph of the section "5. General Discussion and Conclusion".

p.2: Arguments from corpus and processing don't have to be cited to cast doubt on the idealization; infinity cannot be just observed.

We think the corpus and processing arguments are still important and worth mentioning: It implies that we would have little chance of observing center-embedding in animal data. Also, Reviewer 1 seems to appreciate it, so we would want to keep it for such readers unless there is a reason *against* citing the studies.

p.2: I don't understand the sentence "A better empirical argument for the superregularity of human language comes from probabilistic analysis". Better in what respect?

The evaluation here is based on the applicability to animal studies. This is now clarified in the section "2. Preliminaries".

p.2: What is "constant" in the equation? And with this constant, the equation shouldn't be exact in the technical sense.

We replaced the word "constant" with " $-\log p(d)$ ", clarifying what the constant is. (It is just the normalizing constant, and is irrelevant to the model evaluation by Perfors et al.'s study.)

p.3: What does "active" as in "active rules" mean exactly?

We intended to mean "rules with non-zero expected counts of use" by "active rules". To avoid unnecessary confusion, however, we no longer use the term "active" and instead say "used" in the revised draft.

p.3: I don't understand the sentence "Note that regular grammars are a subclass of PCFG and, thus, are included in our hypothesis space". The inclusion of regular languages by context-free languages is not an issue here.

We rephrased the sentence into "... regular grammars are a special case of PCFG that only generate either the left-branching (Figure 1b) or right-branching (Figure 1c) structures." (Section "3. Overview of the Materials & Methods"). The point is that the regular grammars are included in the support of the prior and posterior

distributions over PCFGs, and thus, the analysis of the posterior tells us if regular grammars are really probable hypotheses for the gibbon data as previously suggested even if the hypothesis space is expanded to the class of PCFGs (which turns out not to be the case).

p.3: The phrase "...which is the standard evaluation metric for language models in the field of natural language processing (NLP)" seems unnecessary, simple because this paper is clearly not a NLP paper.

Removed as suggested.

p.3: The length of at least 3, not 4, is minimally required to create non-regular parses.

Strings of the length 3 have two possible parses, and they are left- and right-branching (regular).

p.3: Regarding the sentence "the non-regular parses were significantly more probable than the regular parses", is the "significance" statistical? In general, no statistics are reported in this paper.

The sentence is no longer in the manuscript as we made major revisions in the section "4(a) Posterior Inference of PCFG". On the other hand, we now report the chance-level expected counts of the non-regular parses, and report that the actual expected counts based on the posterior are an order of magnitude greater than this chance level. We still avoided the term "significant" there, though, because we do not have any *distribution* of sampling error to compute p-values etc.

p.4: The amount of training and test data looks unbalanced. Any reason(s)?

This is simply because

- Training was time-consuming: Each iteration of the coordinate ascent algorithm takes an order of cubic time with respect to the length of the data strings. The amount used here was almost the maximum possible given our limited computational resources.
- The size of test data did not have such restrictions, so used the greater amount (recordings over a month). These reasons are mentioned in the last paragraph of the subsection "6(b) Preprocessing".

p.4: Despite the fact that no left-branching parses were Viterbi parses, the log ratio for left-branching was smaller than the one for right-branching. Why?

Before answering to the question, note that

- the log ratio of the expected counts is no longer used as a presentation of the results, and
- the new figure no longer limits the length of the data to 4 or greater, and accordingly there are some Bayes-optimal left-branching parses in the figure.

Now returning to the question, the left-branching parses were never Bayes-optimal (previously called "Viterbi") for the data of the length 4 or greater. However, the posterior probability (or expected counts) of the left-branching parses was greater than the right-branching parses for the *data whose optimal parses were non-regular* (neither left- or right-branching). In other words, the left-branching parses had non-greatest but non-smallest probability for most of the data whereas the right-branching parses were the most probable for some of the data but less probable than the left-branching parses for the others.

p.5: See Berwick (2015) "Mind the Gap" for an argument on grammar compactness.

Thank you for letting us know this work. The revised manuscript cites it in the second paragraph of the section "5. Discussion and Conclusion", where we discuss the evolutionary implications of our study. The reference was

used to point out that the importance of the compactness metric in grammar evaluation has long been emphasized in the field of linguistics.

p.6: Why only first channels of the recordings?

The multi-channel recordings were originally intended for use in sound localization and separation, which were not successful as we reported above. The methods we adopted for the acoustic preprocessing of the gibbon data, on the other hand, were all developed for single-channel data, and we did not have any more effective way of using all the eight channels than simply using the first channel alone.

p.7: How does variable root PCFG make sure that the grammar is consistent in the technical sense?

First of all, allowing variable roots does not itself make any difference from PCFGs with a fixed root. It is equivalent to having a dummy unique root non-terminal, which we name "S" following the convention, and unit production rules from S to the actual non-terminals, $S \rightarrow A$, $S \rightarrow B$, $S \rightarrow C$, and so on. (Note that S never occurs in non-root positions.) We can make small modifications to Chi and Geman's (1998) proof of the consistency of the PCFG induced by the expectation maximization such that it is applicable to the variable-root PCFG. Even though the proof presupposes that there is no unit production rule, the unit production rules from S are only used once per string. Hence, the number of each non-terminal's occurrences in the training data is still upper-bounded, and this makes Chi and Geman's proof applicable to the variable-root PCFG.

On the other hand, our Bayesian posterior induction would not exclude non-consistent PCFGs: i.e., The posterior would assign non-zero probability to non-consistent PCFGs. As far as we know, there is no study that helps us answer to the question how much posterior probability is distributed to the non-consistent PCFGs, and it is out of the scope of the current paper.

For this revision, we decided to limit this discussion to this letter because we think this is a too advanced topic on PCFG to discuss in the main manuscript. Fortunately, Royal Society Open Science publishes this letter along with the main manuscript, so interested readers would still be able to find our discussion. We will move this discussion to the main manuscript, however, if you recommend it.

Responses to Reviewer 3

Thank you for your insightful comments, especially those on our acoustic analysis of gibbon data. Please find our responses to your questions/suggestions below.

Thank you for sharing your work addressing the complexity of grammars to represent non-human song. I found your case study With gibbons to be well written, interesting, and thought provoking. To briefly summarize your work, you build on the methods of Perfors et al. (2011) who hypothesized that the complexity of a grammar (or a generating principle in their case) could be represented by the prior probability of selecting a specific grammar from a family of grammars. The actual process of estimating this probability is tricky and your paper uses variational methods to estimate this. I am not a linguist and am not in a good position to judge how effective this is, but seems clear that other scholars have employed this method and will defer to other reviewers with respect to the appropriateness of whether or not this specific prior estimate is merited.

The idea of balancing between fit to data (which we often call explanatory power in the paper) and model compactness (cf. Occam's razor) is not unique to linguistics: AIC and similar approaches are widely used in the

field of biology. To emphasize this, we cited a wider range of approaches with the same philosophy in the new section "3. Overview of the Materials & Methods" (at the end of the fourth paragraphs there).

While I have heard gibbon calls before, I am not intimately familiar With them (I am a bioacoustian Who works on other taxa). From my memory and a quick review of some of the spectrograms in Clarke et al. (2006)'s well-cited paper on syntax and meaning of pileated gibbons (*Hylobates pileatus*; not *H. agilis* & *H. albibarbis* as used in your study), 100 ms seems reasonably appropriate to me although it is clear that some longer calls will be broken into multiple 100 ms segments. I think it would be appropriate to include a figure that provides examples of the types of segmentations resulting from this type of analysis. The peak selection could also use further clarification.

The revised manuscript now has a more detailed explanation about the motivation behind our segmentation algorithm (the second paragraph of the section "6(b) Preprocessing"). Unlike birdsong syllables, the unit vocal segments of gibbon song are not identifiable from clear silent intervals between them. (This situation would now be clear to you and readers from Figure 4, which we included following your suggestion. The figure shows an example of sound wave and spectrogram of gibbon song. We also visualize the local peaks detected by our algorithm in the figure.) Accordingly, we instead adopted a method developed for analyses of human speech (by the group of Prof. Ghazanfar, an evolutionary speech biologist). The method uses the Hilbert transformed signals and allows us to predict mouth opening actions in human speech, whose acoustic correspondent is syllables.

What does it mean to be "greater than 1.0 of the minimum in a 300 ms search window" (line 52). It is unclear what the units are (counts, dB re. 20 gpa?), making this comparison meaningless.

Sorry, this sentence got confusing (even to us) due to our miscommunication with the English correction service we used. (And we failed to notice it when we double-checked the previous draft after the "correction".)

We rephrased the sentence so that what we intended to mean is clearer: "the difference between the local peak and the minimum in the 300 ms search window was at least 1.0 (Hilbert unit)." The comparison is made in the Hilbert-transformed space, so the unit are what Chandrasekaran et al. (2009) [41] call "Hilbert unit".

The decision to use MFCC and 100 ms frames has several repercussions. On the positive side, it may address the graded vs. discrete call conundrum that we face with non-human calls. On the negative side, long calls will be broken down into low-level acoustic units. When one considers a longer call such as the following: [Drawing] which is roughly sketched from the aforementioned Clarke et paper With a duration Of 1.5 s, there will be 100 ms segments covering things that might be likely to be a single structure, even if one entertains the hypothesis that there may be meaningful subcomponents to this call. This is a bit of a cause of concern to me. I think that this will be likely to introduce some very deterministic structure into the grammars that are generated. It also raises questions (to me at least) about what kind of information was captured and how meaningful it might be.

Just like the segmentation algorithm, we used MFCC following studies on human speech, due to the lack of knowledge about auditory functions of gibbons and "meaningful" acoustic features of gibbon song. (MFCC has also been adopted in the previous studies on primates and some other animals like elephants.) On the other hand, the revised manuscript (in the fourth paragraph of the section "6(b) Preprocessing") also mentions that neural networks have recently started replacing MFCC in the field of computational linguistics, and we emphasize the importance of catching up with such more recent achievements in future studies on gibbon song and other animal vocal activities (just as we do for future grammatical analyses of animal song in the section "5. General Discussion and Conclusion").

For the issue of the frame length, we agree that our 100 ms frames (one frame per syllable) could have failed to capture "meaningful" acoustic information spread over long sound segments, though we do not know what the "meaningful" information is. There are several ways for computational linguists to characterize acoustic information of human speech (syllables) varying in length. Picking up representative frames is an example, though researchers do not agree on any particular method: Some of them just look at the center of the syllables, some use 3-10 equally spaced frames, and some take the average over the entire syllables. A more ambitious approach is to use HMM/RNN to exploit all the frames of speech (e.g., Lee et al., 2015 [17]). Given these various approaches, our choice of the single-frame-per-syllable method was due to a practical reason: We only had limited computational resources and we adopted the simplest acoustic analysis to allocate the resources to the computationally demanding grammatical analysis.

Specific comments:

p1 41 & 52 — cite

The first four sentences are all based on the references [1-3]. We moved the references to the end of the first sentence.

p3 43 — I believe the parse induction experiment is carried out only on the PCFG grammar and the separation is just looking at non-regular vs right-branching rules. If correct, please state, otherwise clarify

You're right, though what was induced was not a single PCFG but rather the posterior *distribution* of PCFGs, which include the regular grammars as a subclass. The analysis of the parses is a way to interpret the posterior and check whether the regular grammars are in fact probable among PCFGs in the posterior as claimed in the previous literature (which turned out not to be the case). (We adopted this indirect analysis of the posterior because it is difficult to directly measure the posterior probability of the regular grammars among PCFGs.) These points are now clarified in the new section "3. Overview of the Materials & Methods".

p 4 50 - Where does the 77 from 52-77 come from?

We tested two ways of counting the expected number of *types* (not tokens) of rules for PCFG/HMM to explain the training data. To evaluate whether each rule was used (1) or not (0), one of the methods applied the step function " ≥ 1 " to the rule's expected token counts while the other applied tanh to make the evaluation continuous rather than categorical. 52 rules were saved by PCFG (in comparison with HMM) when the type counts was computed with the step function, and 77 rules were saved when tanh was used.

The notation "52-77" was inappropriate because it would mean that the two numbers are the lower and upper bounds of the number of rules saved. We revised the section "4(c) Compactness" and the results there would now be clear.

p 4 56 — The conclusion seems a bit overstated to me.

A similar suggestion was made by Reviewer 2, and the last paragraph of the section "5. General Discussion and Conclusion" now includes a caveat that the PCFG induced in our study is just a computationally optimal hypothesis and is not guaranteed to match the actual system behind gibbon song. The paragraph also states that the biological validity of the induced grammar should be assessed in future experimental studies (e.g., by callback experiment) and also that we should consider a broader hypothesis space than the class of PCFGs since gibbon song could be better modeled by super context-free grammars.

Fig 4 — It took me a while to figure out this bar plot. Perhaps change styles (point +/- 95% CI) and make the axis not span 0 to -40?

We changed the visualization and the new figure (Figure 3) shows the distribution (rather than the mean and CI) of the predictive probability with the violin plot.

p 6 40 — A 20 ms window on a signal with $F_s=16$ kHz is 320 samples. This is not a power of 2, so you probably did not use the fast Fourier transform. I suggest changing to discrete Fourier transform (DFT).

Corrected as suggested.

Authors' Responses to the Comments from the Reviewers on RSOS-190139.R1, "Superregular grammars do not provide additional explanatory power but allow for a compact analysis of animal song"

Takashi Morita

Hiroki Koda

Dear Dr. Claudia Wascher, our handling editor of Royal Society Open Science,

We would like to thank you for accepting our manuscript for publication in Royal Society Open Science.

We made minor revisions (summarized below) to address the suggestion raised by Reviewer 3 (and to make a minor correction to our math). A more detailed response to it is found below as well.

Finally, let us express our gratitude again for the editor and reviewers, for their time and constructive comments on our manuscript.

Takashi Morita, on behalf of the coauthor

Summary of the Minor Revisions

The following minor revisions were made in the manuscript.

- The second paragraph of Section "3. Overview of the Materials & Methods" was extended to address Reviewer 3's question why our grammar outputs acoustic vectors rather than discrete sound categories.
- The math about the HDP-HMM in Figure 5b was missing the description about the stopping probability we adopted (the model would generate infinite strings without it), and the final draft now has the correct math there.

Responses to the Reviewer's comments:

Reviewer: 3

Overall, I think you did a nice job addressing most of my concerns. I do not notice some minor errors, such as the omission of Fig 4 which displays as blank (at least with the Chrome and Acrobat PDF viewers) and it would be nice to see what you downsampled to, but that can be inferred.

We are sorry for that problem: The PDF proof of the draft we received at the submission (with Chrome, I think) correctly presented the figure, but we just observed the same problem in the R1 draft accessible from "View Submission" (with Chrome). We also faced the same problem when submitting this revised draft, and it looks that (downloading with) Chrome does something wrong and the figure turns into a blank while Safari seems to work fine in our environment. We will report this issue to the editorial staff.

The only concern of any consequence that is still bothering me is how much the use of acoustic vectors as opposed to symbols influences the analysis. I like the justifications that you have added, but I can't help thinking that it might be more convincing if the same analysis was done on the human infant data base to which you compare. I did not raise this concern the first time I reviewed this as it was not until I read it this time to recognize that this was at the crux of what bothered me about the analysis. I don't know that this analysis is absolutely necessary, I think that you raise interesting ideas and the work stands on its own, but it might be a stronger argument if you could make the apples and apples comparison in future work.

Thank you for raising an important question.

Let us first clarify that our analysis takes the form of a *joint inference* of

1. the discrete categories (corresponding to the "terminals") of the acoustic feature tokens, and
2. the grammar behind the sequential patterns of the categories.

In contrast to our joint approach, these two components have been analyzed typically in separate steps in the previous studies of animal song. However, simulation studies of human language learning have shown that a joint inference of multiple aspects of the target language is more successful than separate learning of the components (because the former can capture correlations among the components existing in the data). Given these findings, we consider that the joint analysis is more reliable than the separate learning of the sound categories and the grammar, and this is the reason why our grammar outputs the acoustic vectors rather than discrete symbols inferred from some preprocessing. The reasoning above is also included in the revised draft at the end of the second paragraph in Section "3. Overview of the Materials & Methods".

On the other hand, we agree that a comparative analysis of human speech data---rather than the text data analyzed by Perfors et al. (2011)---is ideal as you suggested. Unfortunately, the human data were only provided in the text format and we cannot perform the same analysis on it. We are currently working on a comparative study of a wider range of gibbon species, and that should discuss human speech data from multiple languages as well (not just English). And let us finally (re)emphasize that the language (unsupervised) learning from raw speech data is still an active topic of computational linguistics (as represented by the recent Zero Resource Challenge), and we should and will adopt more recent achievements in the field for better comparative studies on human language and animal song.